# Anatomical Schemata Revealed by the Critical View of Safety Approach: A Proposal of the Hellenic Task Force on the Typology of Safe Laparoscopic Cholecystectomy (HETALCHO)

**DOI:** 10.3390/medicina60121968

**Published:** 2024-11-29

**Authors:** Dimitris Papagoras, Gerasimos Douridas, Dimitrios Panagiotou, Konstantinos Toutouzas, Panagis Lykoudis, Alexandros Charalabopoulos, Dimitrios Korkolis, Konstantinos Alexiou, Nikolaos Sikalias, Dimitrios Lytras, Theodosios Papavramidis, Konstantinos Tepetes, Konstantinos Avgerinos, Spyridon Arnaoutos, Konstantinos Stamou, Evangelos Lolis, Dimitrios Zacharoulis, Georgios Zografos, Georgios Glantzounis

**Affiliations:** 1Surgical Department, General Hospital of Trikala, 421 00 Trikala, Greece; dpapagoras@hotmail.com (D.P.); dimpanagiotou@yahoo.gr (D.P.); 2Surgical Clinic Thriasio Hospital Elefsina, 190 18 Elefsina, Greece; makissurgeon@hotmail.com; 31st Propedeutic Department of Surgery, National and Kapodistrian University of Athens, Hippocration General Hospital, 115 27 Athens, Greece; tousur@hotmail.com; 44th Surgical Department, National and Kapodistrian University of Athens, University General Hospital Atttiko, 124 62 Athens, Greece; p.lykoudis@ucl.ac.uk; 51st Surgical Department, National and Kapodistrian University of Athens, Laiko Hospital, 115 27 Athens, Greece; acharalabopoulos@yahoo.com; 6Department of Surgical Oncology, Oncology Hospital Saint Savvas, 115 22 Athens, Greece; dkorkolis_2000@yahoo.com; 7Surgical Department, Kentriki Kliniki, 106 80 Athens, Greece; knegro@otenet.gr; 8Surgical Department, General Hospital Kalamata, 241 00 Kalamata, Greece; niksikalias@aol.com; 92nd Surgical Department, General Hospital Papanikolaou, 570 10 Thessaloniki, Greece; dklyt@yahoo.com; 101st Propedeutic Department of Surgery, Aristotle University of Thessaloniki, AHEPA University Hospital, 546 36 Thessaloniki, Greece; papavramidis@hotmail.com; 11Department of General Surgery, University Hospital Larisa, 413 34 Larisa, Greece; tepetesk@gmail.com (K.T.); zachadim@yahoo.com (D.Z.); 12Surgical Department, General Hospital Bioclinic, 115 24 Athens, Greece; costavger65@gmail.com; 13Surgical Department, General Hospital Sparta, 231 00 Sparta, Greece; ssaadrsu@gmail.com; 14Surgical Department, Privat Hospital Mitera, 151 23 Athens, Greece; kostas@stamou-surgery.gr; 15HPB Unit, Department of Surgery, University Hospital of Ioannina, 455 00 Ioannina, Greece; vaglolis@otenet.gr; 16Surgical Department, General State Hospital Gennimatas, 115 27 Athens, Greece; gnzografos@yahoo.com

**Keywords:** surgical anatomy of cholecystectomy, typology, vascular–biliary injuries, laparoscopic, anatomical schemata, critical view of safety, structured knowledge

## Abstract

*Background and objectives*: Laparoscopic cholecystectomy (LC) is the most commonly performed operation in general surgery in the Western World. Gallbladder surgery, although most of the time simple, always offers the possibility of unpleasant surprises. Despite progress, the incidence of common bile duct injury is 0.2–0.4%, causing devastating implications for the patient and the surgeon. This is mainly due to the failure to identify the normal anatomy properly. The literature review reveals a lack of structured knowledge in the surgical anatomy of cholecystectomy. The aim of this study was to develop a framework with a common anatomical language for safe laparoscopic and open cholecystectomy. *Materials and Methods:* The Hellenic Task Force group on the typology for Safe Laparoscopic Cholecystectomy performed a critical review of the literature on the laparoscopic anatomy of cholecystectomy. The results were compared with those of a clinical study of 279 patients undergoing LC for uncomplicated symptomatic gallstone disease. *Results:* Fourteen elements encountered during LC under the critical view of safety (CVS) approach were determined. The typical vascular–biliary pedicle with one cystic duct distributed laterally (or caudally) and one cystic artery medially (or cranially) lying at any point of the hepatocystic space was found in 66% of the cases studied. Anatomical schemata were formulated corresponding to the norm and four variations. *Conclusions:* The proposed cognitive anatomical schemata summarize simply what one can expect in terms of deviation from the norm. We believe that the synergy between the correct application of the CVS and the structured knowledge of the surgical anatomy in cholecystectomy helps the surgeon to handle non-typical structures safely and to complete the laparoscopic or open cholecystectomy without vascular–biliary injuries.

## 1. Introduction

Anatomy in laparoscopic cholecystectomy is one of the three pillars constituting the cognitive concept of safe laparoscopic cholecystectomy (LC) [1], with the other two pillars being the well-established anatomic landmarks for the orientation of the dissection in a safe “Go” zone [2] and the critical view of safety (CVS), a method of sound identification of the vascular–biliary pedicle [3]. Our working group under the hospice of the Hellenic Society of Surgery and the Hellenic Hepato-pancreato-biliary (HPB) Association (Table 1) has focused, since 2022, on the development of a framework of typology for the safety of laparoscopic cholecystectomy with the purpose to provide the surgeon with a cognitive working plan [4,5] avoiding major vascular–biliary injuries.

These tree pillars are closely interrelated during LC: (1) the landmarks, i.e., the cystic artery lymph node and the sulcus of Rouviere, provide the right orientation for safe dissection [2,6]; (2) the three components (criteria or requirements) of the CVS explain the exact “route” and the purpose of dissection in the relevant surgical spaces [7]; and finally, (3) the anatomy displays the “names” of the structures encountered during the process of dissection, achieving the CVS [8].

These “names” reveal the identity of the anatomic structures encountered during LC and have to be precisely clarified and determined, as stated by Michels in his landmark studies [9,10] about the vascular supply of the liver and the gallbladder. However, the need for a common anatomical language is still unfulfilled [11]. The relevant literature [12,13,14,15,16,17,18,19,20,21,22,23,24,25,26,27,28,29,30,31,32,33,34,35,36,37,38,39,40,41,42,43,44,45,46,47,48,49,50,51] offers incredible data and information, but it is far from what we would call a cognitive system of laparoscopic anatomy knowledge. This is mainly because there is no standard useful to the surgeon’s practical classification of the “infamous” variations [9].

The arrangement in the epicenter of LC concerns the cystic duct and the cystic artery both among themselves and in relation to the gallbladder. This vascular–biliary or gallbladder pedicle displays a typical configuration (i.e., the norm), as represented in the classical figure in Strasberg’s publication [3], and is found in about 75% of cholecystectomies [9,10]. However, in the remaining 25% of cases during dissection, a non-typical vascular–biliary pedicle configuration and/or a third (redundant) anatomic structure can eventually emerge, challenging the CVS requirement of “two and only two structures” [52,53]. For the Surgeons to accomplish the CVS, they have to identify these variations in number or course by names that can only derive from sound anatomical knowledge [54].

In this subject matter, most of the anatomical studies [12,13,14,15,16,17,18,19,20,21,22,23,24,25,26,27,28,29,30,31,32,33,34,35,36,37,38,39,40,41,42,43,44,45,46,47,48,49,50,51], with the exception of Michels’ landmark publications [9,10], express many opinions but do not answer the dilemma of how to manage these variations in course and number eventually encountered during the CVS approach. This is mainly because they (a) largely use heterogeneous definitions without any reference to a methodological basis of support for this anatomical terminology, (b) group the anatomical “variations” either by using anatomical data from cadaveric studies [12,13,14,15,16,17,18,19,20,21] that do not concern the surgical working plane in safe zones or through computer tomography or angiography findings [47,48,49,50,51] that are not easily reproducible surgically, (c) fail to connect the anatomy in LC to the CVS approach and finally (d) omit practical considerations and data on anatomic structures that may be present and eventually course under the cystic plate in the liver bed, an essential component and surgical working space in the CVS concept.

Data point out that the anatomical identification of structures during laparoscopic cholecystectomy, even years after the introduction of the CVS, is a troublesome task for surgeons [55,56]. In our own questionnaire at the annual workshop on the typology of safe laparoscopic cholecystectomy, 42 of the 80 trainees answered that the most significant difficulty they face in laparoscopic cholecystectomy is the identification of anatomic structures. We believe this is not only related to the incomplete understanding of the three components of the CVS [57] but also to the lack of structured knowledge in the surgical anatomy of cholecystectomy. The enormous importance of structured anatomy knowledge in LC is summarized in one of the six points of the strategy for safe cholecystectomy in the phrase “understand the potential of aberrant anatomy” [58], which is also underlined in the guidelines of the European Association for Endoscopic Surgery (EAES) [59].

The aim of our study is to generate data through a critical literature review and comparison with the results of our own study, which will allow us to understand aberrant (variant, non-typical) anatomy in LC and to form a common anatomical language combined with cognitive anatomical schemata that can be used as an anatomical map, helping the surgeon to handle with confidence any variation that occurs in the process of achieving the CVS.

## 2. Materials and Methods

To understand this aberrant anatomy in LC, our working group followed investigative pathways in order to (1) rule out differences between the anatomical terms “norm” and “variations” in terms of frequency and configuration of the gallbladder pedicle, (2) clarify the names (i.e., the identity) and the prevalence of the anatomic elements that will or can be encountered during LC in the process of achieving the CVS, (3) determine the frequency of the typical and the non-typical configurations of the gallbladder pedicle (i.e., variations in number and in position) in LC, (4) delineate the anatomical identity of the supernumerary (“third element”) and the heterotopic (non-typical distribution) course of a “variant” and the frequency by which these non-typical structures are eventually found in an effort to achieve the CVS and (5) group these anatomic elements into understandable anatomical schemata [60,61] of typical and non-typical arrangements of the vascular–biliary pedicle.

We used the two benchmark publications by Michels [9,10] as a glossary of anatomy to clarify and establish a terminology of the anatomic elements relevant to the procedure of cholecystectomy under the dictum of the CVS. We reviewed the SAGES manual for safe cholecystectomy [58], the published anatomical studies in laparoscopic cholecystectomy [22,23,24,25,26,27,28,29,30,31,32,33,34,35,36,37], review articles [38,39,40,41,42,43,44,45,46], studies in cadavers [12,13,14,15,16,17,18,19,20,21], angiographic imaging studies [47,48,49,50,51] and open cholecystectomy studies [52,60] to sum up the anatomical terms of the structures, the frequency of their occurrence, the rational of their grouping described in each publication. We discerned the “norm”, the “variations”, and their subcategories according to Kachlik et al. [62]. The term “norm” is equal to “usual, typical, standard form, or a regular pattern”. The term “variation” refers to any “variety from the standard form”. Regarding the prevalence, a variation is characterized as “frequent” if it occurs at a frequency > 10% and up to 50% of the cases. “Infrequent variations” express an occurrence rate of 1–10%, whereas “rare variations” are present in <1% of cases, and finally, “sporadic variations” represent case reports [62,63,64].

We underlined that any variation in laparoscopic anatomy of cholecystectomy adherent to the CVS approach concerns two subtypes, regardless of their frequency: one refers to a supernumerary structure, i.e., a third element (variation in number), and the other to the non-typical course of a structure (variation in course), with the notion that these subtypes of variations can coexist [40]. The variation in course creation is synonymous with the “transposition of the gallbladder pedicle” [33]. We discerned two working spaces in LC that adhere to the CVS approach: the hepatocystic space and the cystic plate. The hepatocystic space has to be cleared “of the fat, fibrous and areolar tissue” (the 1st component of the CVS) [3]. We presume that the distal boundary of this space is not always the cystic duct because there are cases with a transposition of the gallbladder pedicle in which the cystic artery is coursing lower (caudally) than the cystic duct. The hepatocystic space contains the cystic duct, the cystic artery, the arteries of Calot (small branches coursing between the cystic duct and the cystic artery), the cystic artery lymph node in the lower border of the gallbladder wall, the right hepatic artery, and eventually an aberrant, right hepatic duct and an accessory or replaced right hepatic artery [41]. The cystic plate is a thin layer of fibrous fat and areolar tissue covering the liver bed of the gallbladder [65,66,67,68,69]. Under this layer, the following structures are distributed: the subserosal ducts of Haberland [9], the deep cystic artery either as a branch of the cystic artery or a separate artery [38,39,41,46] and branches of the middle hepatic vein [9]. The cystic plate has to be left intact on the liver bed during the elevation of the lower part of the gallbladder, “which has to be dissected off the liver bed to expose the lowest part [of the posterior surface] of the gallbladder” (the 2nd component of the CVS) [70]. This surgical maneuver ensures that a possibly present aberrant right hepatic duct is protected from injury [3].

The meticulous and complete dissection in these two working areas converge in the 3rd CVS requirement: “There should only be two structures seen entering the gallbladder, and the bottom of the liver bed should be visible” [3]. Only after the fulfilment of these three components, “the surgeon has achieved the CVS and the cystic structures may be occluded because they have been conclusively identified” [3]. Taking these into consideration, we defined anatomic structures as “targets”, referring to constant elements which have to be dissected, conclusively identified and occluded above the level of Rouviere’s sulcus [71] in the so-called safety or “Go zone” [1,2,72], and “non-targets”, representing anatomic elements not encountered unless they invade this “Go zone”, necessitating their recognition and protection from injury by careful dissection (Figure 1). We defined any supernumerary anatomic element exposed during the process of achieving the CVS, besides the cystic duct and the cystic artery, as a third structure. This third anatomic element could be either an artery, i.e., a double cystic artery, a right hepatic artery with a caterpillar hump, an aberrant artery (accessory or replaced right hepatic artery) or an aberrant right hepatic duct. We defined as a heterotopic structure any distributional pattern of the cystic artery that deviates from the typical configuration of the gallbladder pedicle as depicted in the classical scheme in the publication above by Strasberg et al. [3]. These aforementioned definitions drove us to (a) conceptualize a cognitive diagram of anatomy in LC in the context of the CVS (Figure 1) and (b) clear up and include in a nomenclature all the anatomic elements that can be encountered during LC (Table 2).

Finally, we reviewed 279 videos of LC performed under the dictum of the CVS with the purpose of creating our own point of reference regarding the anatomic elements in terms of their frequency and their variation prevalence and pattern. All procedures were video-recorded with a KARL STORZ IMAGE1 S™ Rubina^®^ camera platform that provides native 4K UHD resolution. The procedures were performed under the premise of the CVS by two experienced surgeons and included only LC for symptomatic cholelithiasis with no acute or chronic inflammation with fibrosis, i.e., biliary inflammatory fusion.

## 3. Results

Anatomic structures: Prevalence.

The anatomic structures and the identity of the third element found in our material are shown in Table 3.

The norm: The typical gallbladder pedicle.

The typical vascular–biliary duct pedicle with one cystic duct distributed laterally (or caudally) and one cystic artery medially (or cranially) lying at any point of the hepatocystic space was found in 184 cases (66.3%) (Table 3, Figure 2). By adding the 49 (17.56%) cases with low (early) or ultra-low division of the cystic artery (Table 3, Figure 3), the total number of typical gallbladder pedicles found in our material was 233 (83.86%). The cystic duct was encountered in all of the 279 cases.

Non-typical gallbladder pedicle configuration: The third structure (i.e., variation in number) and the transposition of the cystic duct and cystic artery (i.e., variation in position)

The supernumerary-third element was found to have an arterial identity in all except 1 of the 42 (15.05%) cases in the 279 videos. Specifically, 36 (12.9%) cases corresponded to the dual (double) cystic artery (Figure 4) and 5 cases (1.79%) to a right hepatic artery with a caterpillar hump (Figure 5). In all of the 36 cases with double cystic artery, the course of the additional vessel had a clear direction towards the gallbladder. The five cases with the caterpillar hump hepatic artery formed the “unwanted third wheel” [73] that invaded the safety or Go zone and was carefully dissected, identified and protected away from the cystic duct and the gallbladder. One cystic artery in two cases and two cystic arteries in the remaining three cases were found to originate from the convex surface of the hepatic artery with the caterpillar hump. The only non-vascular (i.e., non-arterial) recognized third element corresponded to an aberrant right hepatic duct with low confluence to the cystic and left hepatic duct, forming a triad (trifurcation) of extrahepatic ducts (Figure 6). In our case, the aberrant right hepatic duct was in close proximity to the cystic duct. The surgeon followed the basic principles for achieving the CVS, with a careful dissection close to the gallbladder wall and above the aberrant right hepatic duct that was dissected away from the cystic duct, protecting this vital structure without the need for cholangiography.

The non-typical topographical interrelation between the two structures of the pedicle (i.e., variation in course) was exclusively the result of an atypical distribution (heterotopic course) of the cystic artery in 24 cases (8.6%). The cystic artery in 14 (5.01%) cases crossed anteriorly (Figure 7), in 4 cases (1.43%) hooked or twisted around the cystic duct (Figure 8) and in 6 (2.15%) coursed caudal to (i.e., lying lower than) the cystic duct (Figure 9). These non-typical distributions of the cystic artery in relation to the cystic duct were associated with a supernumerary cystic artery in 6 out of the 14 cases of the crossing the cystic duct artery, 3 out of the 4 cases of the artery hooking of the cystic duct and 5 out of 6 cases of the cystic artery lying lower than the cystic duct.

The low-lying cystic artery (Figure 9) also represents a point of confusion. This artery is a course in the hepatocystic space’s outer (caudal) border, causing the transposition of the gallbladder pedicle [33] with the cystic duct lying medially in the hepatocystic space. Some studies describe it as “anterior” [30] and others as an “inferior” cystic artery [39].

The deep cystic artery, as a branch of the cystic artery coursing beyond the confines of the gallbladder wall on the liver bed, was found in a percentage of 12.54% (Figure 10).

We did not find any case with a duplication or absence of the cystic duct, underlining the fact that this aberration of the cystic duct is sporadic. Structures classified by us as “various” were detected at a frequency of 1.43% concerning elements that are not primary targets of dissection, namely, one case with aberrant right hepatic artery, two cases with subvesical ducts of Haberland [9] and one case with middle hepatic artery on the liver bed. The identification of subvesical ducts of Haberland in both cases was made after their injury due to violation of the cystic plate with concomitant cholorrhea, which was repaired by placing a clip. No other structure was violated or injured in the remaining cholecystectomies. No intraoperative cholangiography was performed.

Notice that the ultra-low division can represent a double cystic artery. In both instances, the ligation of the two branches should be performed separately, in contrast to the low division of the cystic artery, which can be safely clipped before the division point.

## 4. Discussion

Gallbladder surgery, although most of the times simple, always offers the possibility of unpleasant surprises.

Surgical anatomy is a dynamic process of categorizing anatomic elements after they have been made visible through dissection during surgery [54]. This explains the dialectical relationship between laparoscopic anatomy and the CVS approach: we need to know the identity of the supernumerary or heterotopic element that is encountered by dissection during the phase of achieving the CVS. For the few square centimeters of the hepatocystic space, there are an abundance of anatomical data and their categorizations, which, until today, have not been able to create a single anatomical language for one of the most frequently performed procedures in general surgery. This “blind spot” in anatomical knowledge can interfere with the interpretation of the CVS and compromise the safety of LC. Confronted with this deficit in structured anatomical knowledge regarding LC, we first clarified the anatomical names (i.e., terms) of structures that can or will be encountered during LC coherent to the CVS approach (Table 2). For this purpose, we relied on the landmark studies by Michels. We created a cognitive anatomical diagram or platform (Figure 1) that includes all the key facts that connect the concept of the CVS with the laparoscopic anatomy of cholecystectomy. We, thereby, formed the methodological basis of an anatomical nomenclature regarding LC performed under the prism of the CVS (Table 2). Our results show, in line with some other published studies, that an atypical gallbladder pedicle, either in terms of a supernumerary “third” element or of a non-typical distribution of the cystic artery in relation to the cystic duct, has a prevalence rate of 16.14%. This is much lower than the reported rate of arterial variation in the literature (ranging between 25 and 50%) [9,33]. The explanation of this is that we (a) excluded variations based on the vessel of origin of the cystic artery, something impractical and not reproducible in daily praxis, and (b) do not regard the low division of the cystic artery as a variant form in terms of a double cystic artery, which can give the impression of a supernumerary element.

Nevertheless, this 16.14% of the non-typical configuration found in our material indicates that variations in LC should be characterized as frequent. Be that as it may, we believe that the key to “understanding aberrant anatomy” as a prerequisite for safe LC [58] lies not in any impractical and unproductive memorization of the prevalence of “unusual” or aberrant (i.e., variational) structures but in the simple and rational grouping of them in topographical and numerical arrangements that these structures have with respect to each other, as well as in the awareness or realization that these “variations” are constant variables, i.e., they are an integral part of everyday surgical practice. Based on the structures described precisely by Michels (Table 2) and found in our material (Table 3), we formulated an anatomical schema corresponding to the norm and four variations schemata (Figure 11). We also felt that it was necessary to clarify the relevant anatomical terms that created a sense of confusion. Typical examples of such misleading terms described in the literature are (a) the “cystic artery originating directly from the liver parenchyma” [27], (b) the “recurrent cystic artery” [24,25], (c) the “cystic artery originating from the gastroduodenal artery” [28] and (d) the “aberrant right hepatic artery with caterpillar hump” [74]. “Variants” (a) and (b) are not described by Michels, nor have we encountered them. We consider that they concern the branch of the deep cystic artery—an anatomic element almost ignored by most of the authors—that courses on the liver bed beneath the cystic plate sheath and which gives tiny branches towards the gallbladder wall [9]. Variant (c) represents the low-lying cystic artery in relation to the cystic duct. This artery is also described as “inferior–lateral” and anterior [34]. Finally, the caterpillar-like hump of the right hepatic artery is misleadingly described as “aberrant right hepatic artery” [73,74]. An aberrant right hepatic artery represents a different anatomic entity and should not be confused with the tortuous course of a right hepatic artery.

The low division of the cystic artery in a superficial and deep branch (Figure 3) visualized after dissection in the hepatocystic space is underreported in studies. We found this configuration in 49 (17.56%) of our cases. This low or ultra-low branching of the cystic artery should not be regarded as a supernumerary/third element or as a non-typical configuration of the gallbladder pedicle. The low-lying cystic artery (Figure 9) also represents a point of confusion. This artery courses in the outer (caudal) border of the hepatocystic space, causing the transposition of the gallbladder pedicle [33] with the cystic duct lying medially in the hepatocystic space. Some studies describe it as an “anterior” [30] and others as an “inferior” cystic artery [39].

After clarifying the anatomical definitions (Table 2) and analyzing our data (Table 3), we grouped the anatomic elements into typical and non-typical arrangements (Table 4). We formed simple anatomical schemata, as shown in Figure 11. These schemata can be easily recalled because they are linked to the reasoning of the CVS, which is the identification of topographical and numerical association of tubular structures in relation to the gallbladder and the cystic plate. We have demonstrated that the prevalence rates of the variations in number (third element) and in position are 15.05% and 8.6%, respectively. With the exception of one case, the third element was a cystic artery or cystic artery branch. These data help the surgeon to understand that despite the detection of the additional third element combined or not with a heterotopic structure, it is ultimately about two and only two structures: an artery that may be non-typical in course or supernumerary and a cystic duct. Careful attainment and subsequent assessment of CVS in all three components will also prevent the ligation or injury of an aberrant right hepatic duct, which is much less common than vascular variants.

We found only 1 (0.35%) out of 279 cases of the aberrant right hepatic duct that formed a biliary triad (trifurcation) after coursing at a close distance from the cystic duct. We emphasize that by dissecting in a correct plan in the well-described safety or Go zones (1, 2) and following the principles of safe cholecystectomy focusing on achieving the CVS [72], the surgeon either does not notice this variant or, when this non-target structure “invades” the safe zone, will identify it conclusively under the condition that severe acute and chronic cholecystitis are absent. Laparoscopic anatomy in cholecystectomy is not distorted by any variation but mainly by the prevalence of acute and chronic inflammatory conditions in and around the hepatocystic space, which should prohibit any attempt to expose the anatomy because of the danger of injuring vital anatomic elements [75].

We also focused our attention on the surgical working space that refers to the detachment of the cystic plate from the gallbladder wall (the second prerequisite of the CVS), because beneath the cystic plate can course the deep cystic artery, a structure that has not been sought as a separate element in any of the relevant studies of laparoscopic anatomy [22,23,24,25,26,27,28,29,30,31,32,33,34,35,36,37]. We detected this artery coursing beyond the confines of the gallbladder wall on the liver bed in a percentage of 12.54%. The deep cystic artery has been incorrectly described as an aberrant right hepatic artery [76], a recurrent cystic artery [25,27,53] or even a vessel without anatomical identification [37]. Beneath the cystic plate, there may additionally be tiny twigs of subvesical bile ducts of Haberland, as described by Michels. These “filament-like bile ducts emerge from the liver substance in the gallbladder bed” [9] and are mistakenly named “Luschka’s ducts” in the surgical literature, a term that should be abandoned [58,77]. These subvesical ducts and the middle right hepatic veins should be protected from injury that may occur when the cystic plate is not kept intact covering the liver bed. We encountered these two structures at rates of 0.71% and 0.35%, respectively.

We included the cystic artery lymph node [78] and the arteries to the cystic duct (Calot’s arteries) [29] in the elements searched for in our material because they reflect important aspects of safe LC. The arteries of Calot should be carefully cauterized after making them visible, while the cystic artery lymph node, a safety landmark, should, whenever possible, be carefully dissected off the cystic artery without an attempt to remove it. Rouviere’ sulcus—described by Michels as a hepatic fissure [9,10]—has been regarded since the publication by Hugh [71] as a reliable landmark for the demarcation of a safety dissection zone. The prevalence of all these three elements in our material ensures that they can be valued as a “norm” in the LC procedure.

### Limitations of the Study

The patients included in the study had uncomplicated gallstone disease, while patients with inflammation, fibrosis or porcelain gallbladder were excluded. This limits the generalizability of the findings to more complex cases with severe acute or chronic cholecystitis, which often poses significant challenges in anatomy visualization. However, even in cases of inflammation or fibrosis, our study offers a framework of surgical anatomy for safe cholecystectomy. In these complex cases, the preoperative use of 3D angiography [49,50] and MRCP or intra-operative cholangiography could be useful in identifying variations in the cystic artery and biliary tree anatomy. Another limitation could be the potential complexity in clinical application: the cognitive framework may be challenging for some practitioners to implement, especially in high-pressure or emergency situations.

## 5. Conclusions

In summary, based on the nomenclature proposed by Michels [9], we determined 14 elements (Table 2) that will or can be encountered during LC in the context of the CVS approach (Figure 1). We defined the variations either in number (third element) or in the course (heterotopic cystic artery) as non-typical gallbladder pedicle configurations. In our review of 279 videos of routine laparoscopic cholecystectomies without severe acute inflammation and without chronic fibrosis, we found the existence of a third structure in 42 (15.04%) and a heterotopic course of the cystic artery in 24 (8.6%) of our cases. In all but one case, the additional third structure involved a supernumerary branch of the cystic artery. We also believe that we resolved certain discrepancies in the anatomical terminology, re-establishing the nomenclature defined by Michels [9,10]. Understanding surgical anatomy in cholecystectomy represents the sine qua non condition to name decisively and correctly identify a non-typical structure. The proposed cognitive anatomical schemata summarize simply what one can expect regarding deviation from the norm (Figure 11, Table 4). The synergy between the correct application of the CVS, by which a third element or a heterotopic structure (or both) are made visible, and the structured knowledge of anatomy in LC gives the surgeon the ability to understand how to handle correctly and with confidence these non-typical structures, completing the LC without major vascular–biliary injuries.

## Figures and Tables

**Figure 1 medicina-60-01968-f001:**
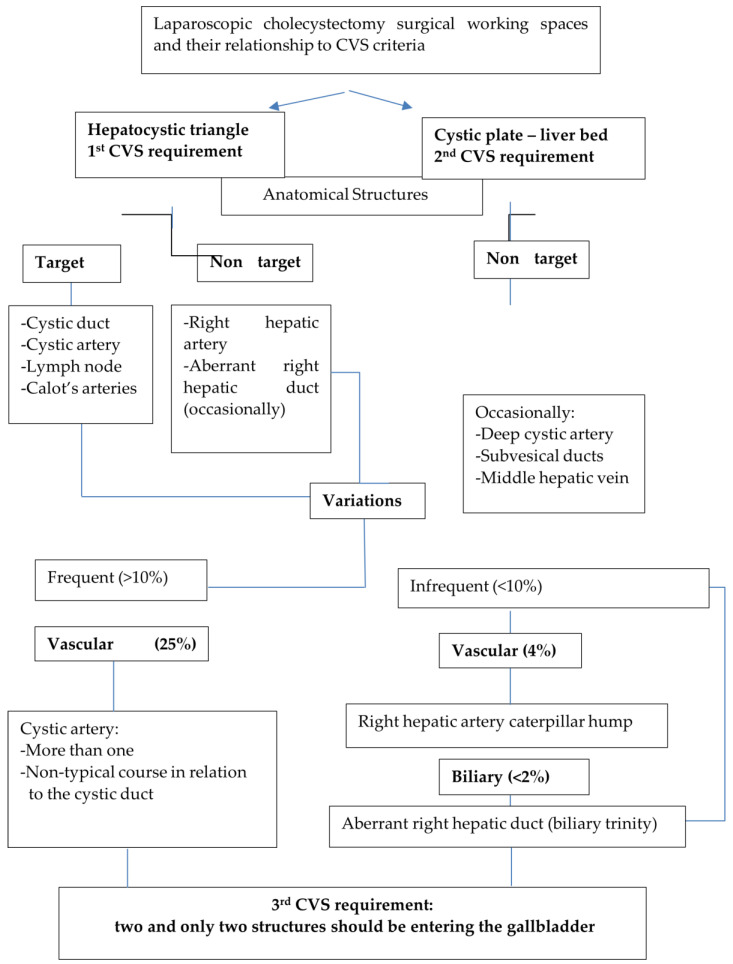
Cognitive framework for laparoscopic cholecystectomy surgical anatomy according to working space and CVS criteria.

**Figure 2 medicina-60-01968-f002:**
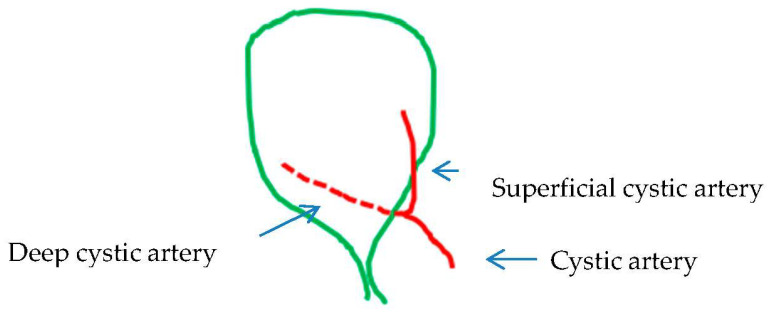
Vascular–biliary pedicle: the norm.

**Figure 3 medicina-60-01968-f003:**
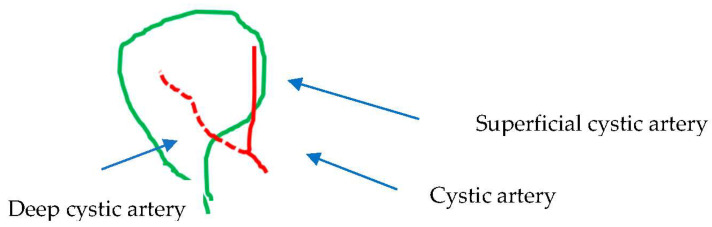
Vascular–biliary pedicle: the norm with low (or early) division of the cystic artery.

**Figure 4 medicina-60-01968-f004:**
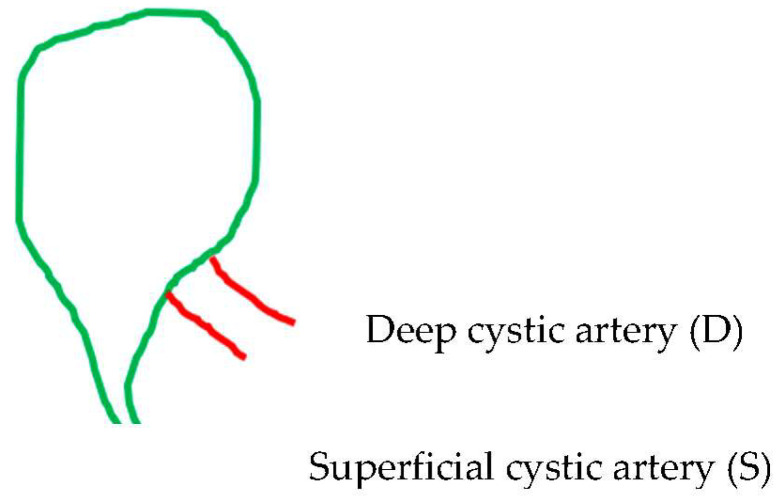
Double (dual) cystic artery: one superficial (S) and one deep (D) cystic artery.

**Figure 5 medicina-60-01968-f005:**
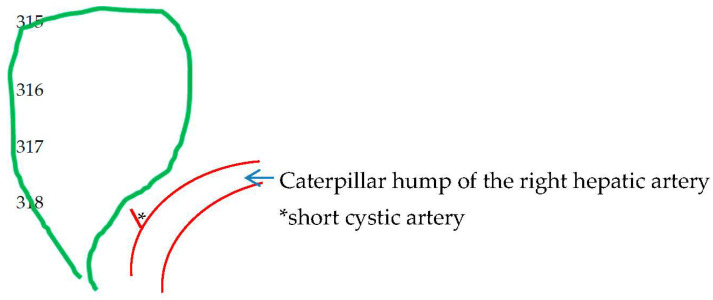
Caterpillar hump of the right hepatic artery with cystic artery, which most of the time is short in length and possibly more than one in number.

**Figure 6 medicina-60-01968-f006:**
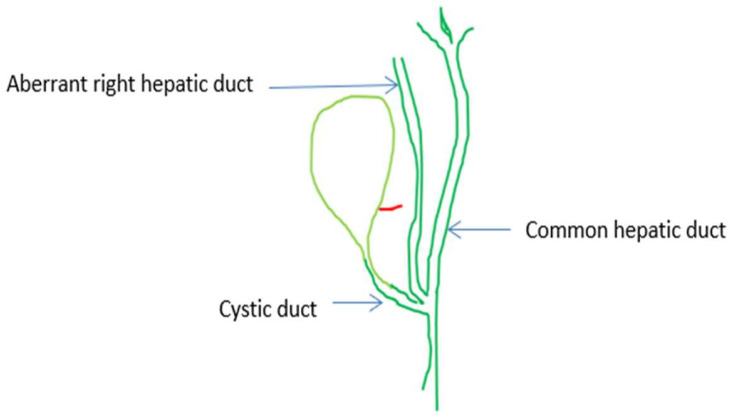
The aberrant right hepatic duct and the cystic duct join the common hepatic duct at the same level, forming the extrahepatic trifurcation [65] (red:cystic artery).

**Figure 7 medicina-60-01968-f007:**
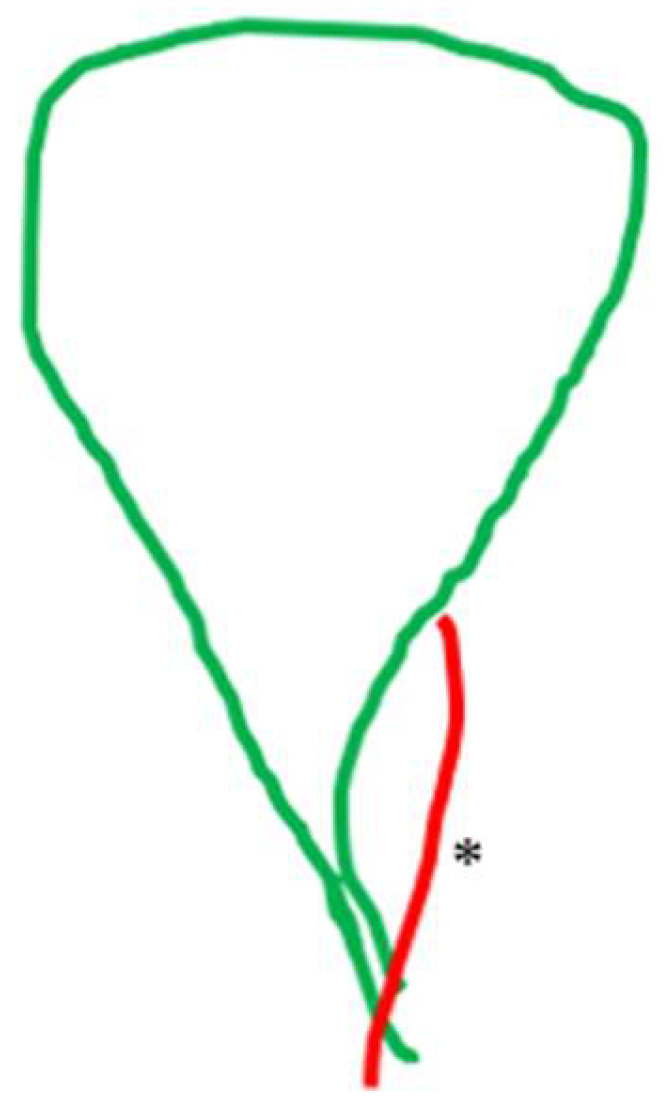
Cystic artery (*) crossing the cystic duct.

**Figure 8 medicina-60-01968-f008:**
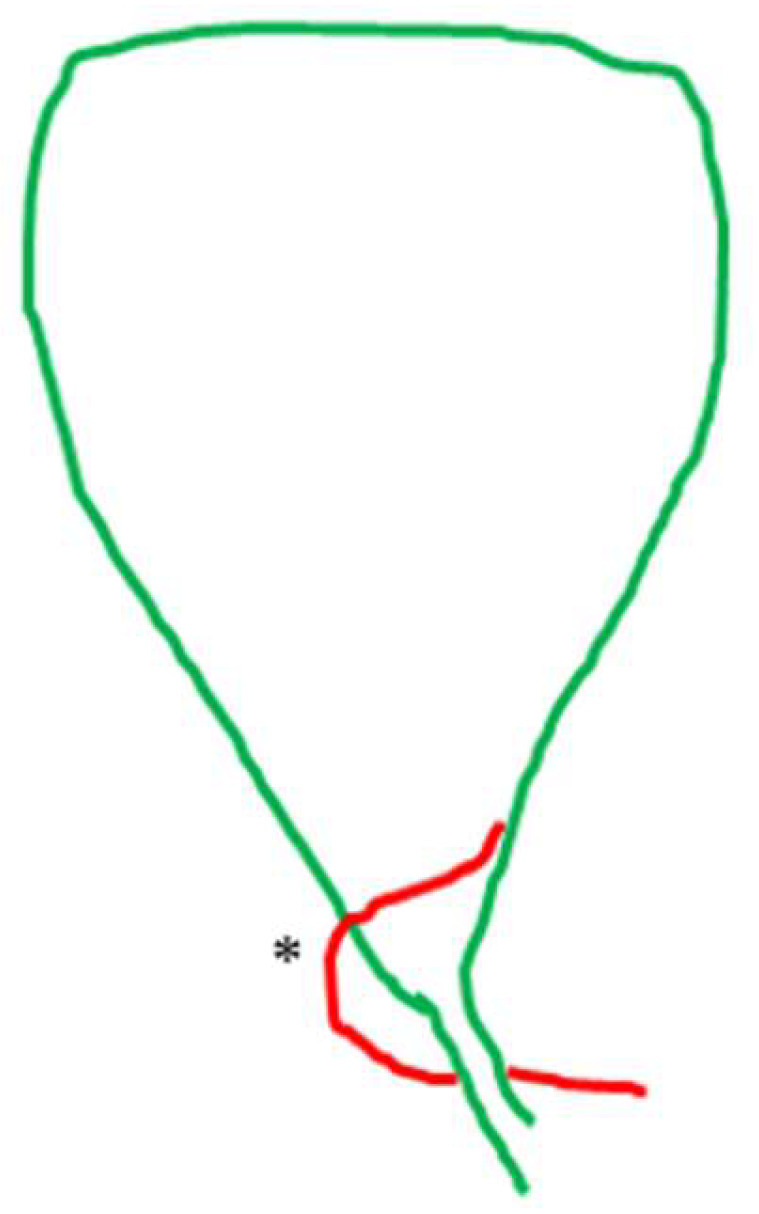
Cystic artery (*) hooking or swinging around the cystic duct.

**Figure 9 medicina-60-01968-f009:**
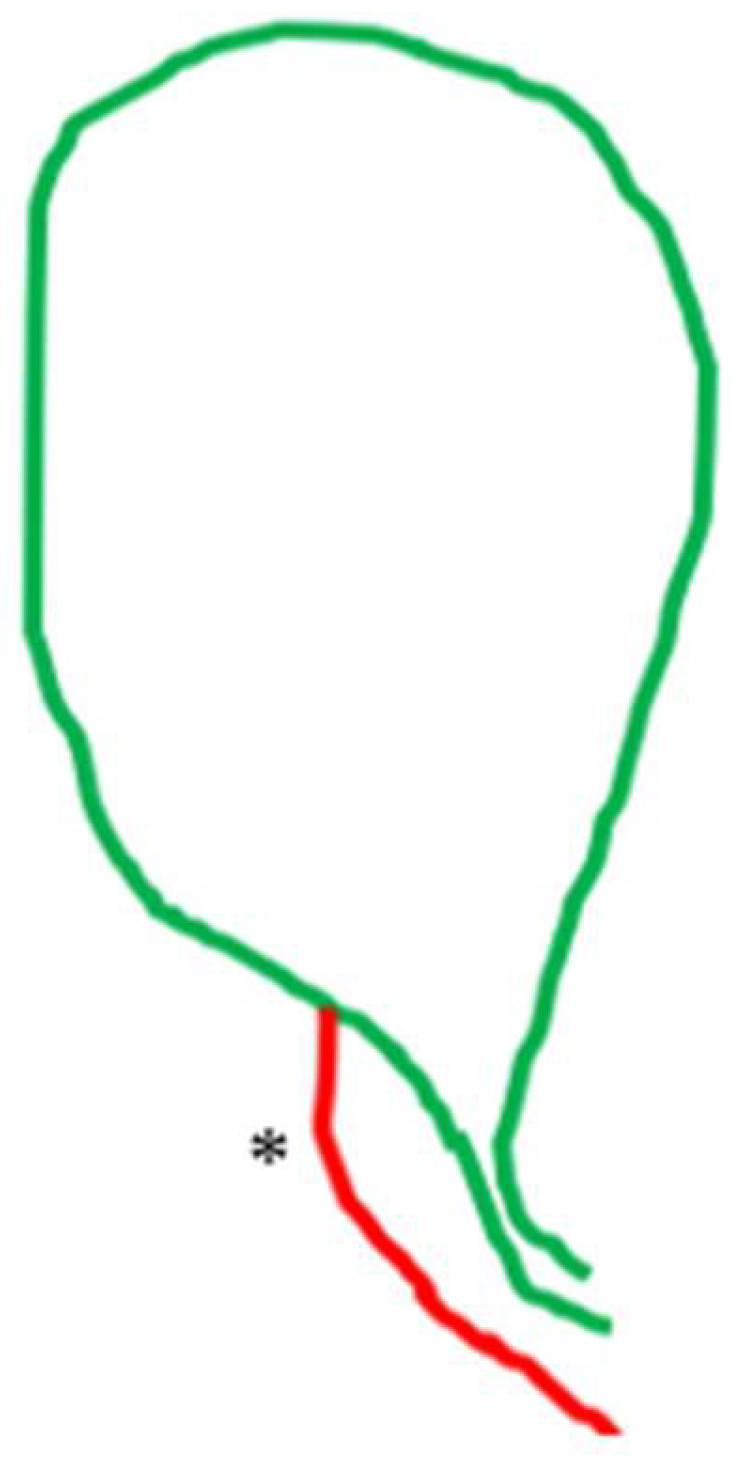
Cystic artery (*) lying lower than the cystic duct. Notice that in this case, the outer border of the hepatocystic space is the cystic artery and not the cystic duct.

**Figure 10 medicina-60-01968-f010:**
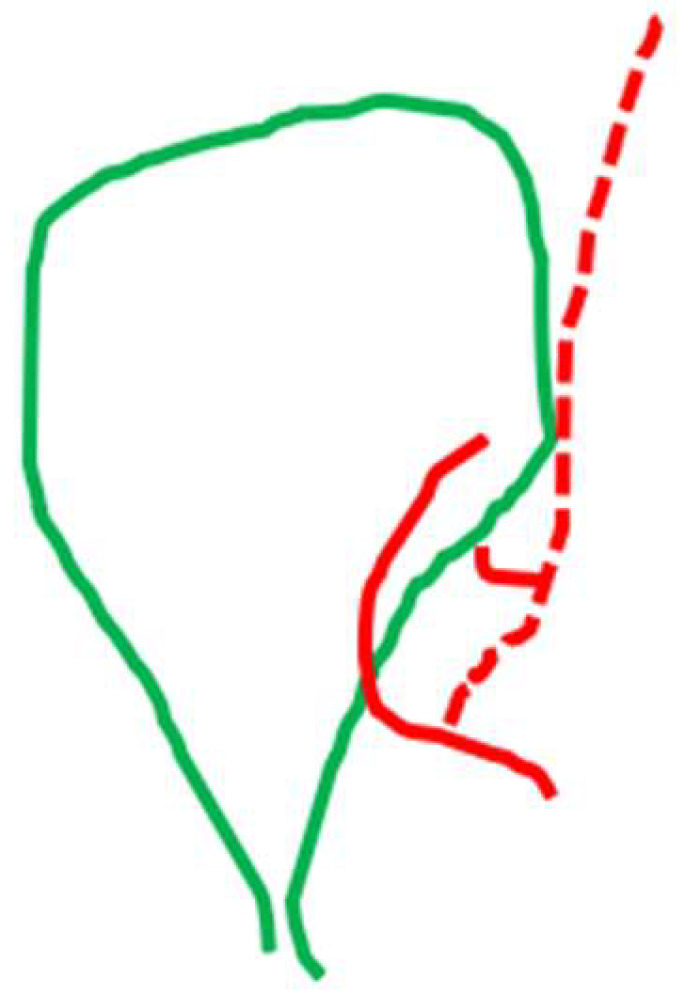
Deep cystic artery (dotted line) as a branch of the cystic artery coursing outside the confine of the gallbladder, giving a small branch to the gallbladder wall and then coursing towards the liver bed.

**Figure 11 medicina-60-01968-f011:**
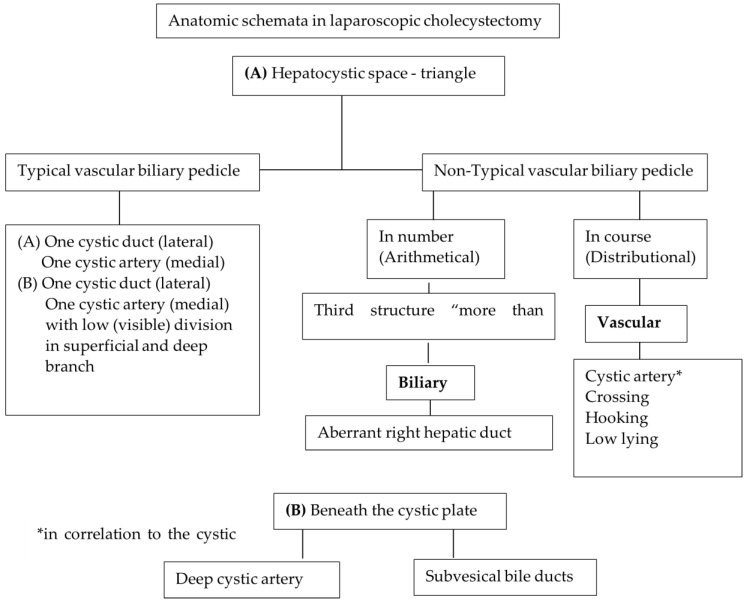
Anatomical schemata in laparoscopic cholecystectomy related to the hepatocystic space and cystic plate.

**Table 1 medicina-60-01968-t001:** Hellenic Task Force for Safe Laparoscopic Cholecystectomy broken down by leaders/working group members on individual topics.

Study Topic	Lead	Co-Lead	Working Group Members
Definition and delineation of typology of safe cholecystectomy as cognitive system	D. Papagoras	D. Zacharoulis	K. Alexiou, N. Sikalias, D. Lytras and K. Stamou
Laparoscopic anatomy of cholecystectomy: definition of variations, anatomical schemata, anatomic landmarks, and Go and No-Go zones	D. Papagoras	G. Glantzounis	D. Panagiotou, D. Giakoustidis, D. Symeonidis, G. Chrιstodoulidis and N. Ropotinos
Understanding the rationale of critical view of safety (CVS), error traps and contraindications to attempting safe CVS, Go and No-Go rules in terms of safe dissection	P. Lykoudis	A. Charalabopoulos	K. Avgerinos, K. Manes, S. Sotirianakos and D. Manatakis
Definition of acute cholecystitis, chronic cholecystitis and acute-on-chronic cholecystitis	K. Toutouzas	D. Stefanidis	K. Fotiadis, A. Kyriakidis and A. Chamzin
Definition, rationale, indications, techniques, complications and follow-up of bail-out strategies	D. Korkolis	G. Zografos	G. Ayomammitis, A. Ninos, C. Rammos and K. Garoufas
Tenets of typology for safe cholecystectomy	S. Arnaoutos	T. Papavramidis	M. Efthimiou, I. Paraskevopoulos and I. Balogiannis
Typology of management of bile duct injuries and legal implications	G. Douridas	A. Vezakis	P. Kokoropoulos, H. Zerbini and E. Iliopoulos

**Table 2 medicina-60-01968-t002:** Nomenclature of anatomic elements encountered during LC adherent to the CVS approach.

Elements That Can Be Encountered During the Achievement of the CVS	
A. Ductal anatomic structures	1. Cystic duct 2. Aberrant right hepatic duct 3. Common bile duct (visible with no dissection)
B. Arterial anatomic structures	4. Cystic artery coursing normally 5. Low cystic artery division 6. Deep cystic artery with a separate course coursing on the liver bed, outside the confine of the gallbladder wall beneath or inside the cystic plate 7. Double cystic artery 8. Cystic artery crossing the cystic duct 9. Cystic artery swinging or hooking around the cystic duct 10. Low-lying cystic artery, i.e., cystic artery caudal to the cystic duct 11. Cystic artery originating from the right hepatic artery with caterpillar turn or Moynihan’s hump 12. Arteries of Calot
C. Anatomic landmarks	13.Cystic artery lymph node 14. Rouviere’s sulcus
D. Other structures	15. Any extra anatomic element not included in this list

**Table 3 medicina-60-01968-t003:** Frequency of anatomic elements found in 279 laparoscopic cholecystectomy videos.

Frequency of Anatomic Elements Encountered	Presence Confirmed
Single cystic duct	279 (100%)
Rouviere’s sulcus (open, slit or scar)	252 (90.32%)
Cystic artery lymph node	250 (89.6%)
Calot’s artery	218 (78.13%)
Single cystic artery	184 (66.3%)
Low division of cystic artery in superficial and deep cystic branch	49 (17.56%)
Double cystic artery	36 (12.9%)
Deep cystic artery on liver bed	35 (12.54%)
Common bile or hepatic duct (visible)	26 (9.31%)
Cystic artery crossing cystic duct	14 (5.01%)
Low-lying cystic artery (inferior to cystic duct)	6 (2.15%)
Cystic artery deriving from caterpillar hump right hepatic artery	5 (1.79%)
Cystic artery swinging around cystic duct	4 (1.43%)
Aberrant right hepatic duct	1 (0.35%)
Other elements (not included in list)	
- Subvesical bile duct (injured to violation of cystic plate plane)	2 (0.71%)
- Aberrant right hepatic artery	1 (0.35%)
- Middle hepatic vein in liver bed	1 (0.35%)

**Table 4 medicina-60-01968-t004:** Grouping of typical versus non-typical (or variant or aberrant) anatomy encountered during laparoscopic cholecystectomy.

Forms of Typical Versus Non-Typical (Aberrant or Variant) Anatomy	*n* = 279
Typical gallbladder pedicle:	184 (66.3%)
Typical gallbladder pedicle with low branching of cystic artery	49 (17.56%)
Non-typical gallbladder pedicle:	
Variations in position:	
Cystic artery crossing cystic duct	14 (5.01%)
Cystic artery hooking (swinging around) cystic duct	4 (1.43%)
Cystic artery “in front” of cystic duct/low-lying cystic artery	6 (2.15%)
Variation in number:	
Double or dual cystic artery	36 (12.9%)
Caterpillar hump of right hepatic artery (“unwanted third wheel”)	5 (1.79%)
Deep cystic artery beyond confines of gallbladder	35 (12.54%)
Aberrant right hepatic duct	1 (0.35%)

## Data Availability

Data of this study (e.g., videos) are available upon request.

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
