# Peer review of "Anatomical Schemata Revealed by the Critical View of Safety Approach: A Proposal of the Hellenic Task Force on the Typology of Safe Laparoscopic Cholecystectomy (HETALCHO)"

_medicina, 2024, doi:10.3390/medicina60121968_

Round 1

Reviewer 1 Report

Comments and Suggestions for Authors

Dear Colleagues! I have read the article with pleasure. I have several suggestions for you to improve the perception of the content by readers. It is advisable to revise the title, since it contains repetitions (laparoscopic, safe), but the anatomical zone of the typology is not indicated. In the "Keywords" "cholecystectomy" is repeated twice. The phrase "anatomy of cholecystectomy" used in the "Keywords" and the text of the article seems incorrect to me. I can suggest, including for the title: 1) "surgical anatomy for safe LC" ; 2) anatomy of " gallbladder pedicle" (line 158) or "hepatocystic space and the cystic plate" (line 159) or "hepatocystic triangle" (line 271); 3) "LC surgical working spaces" (line 209). There is a typo in line 152: instead of the number 65 it should be 64. After "Conclusions" you have not filled in lines 490 and 493-509. These are my comments that are not fundamental. I would like to see your research data in international guidelines on biliary surgery.

Author Response

  • Comment 1: It is advisable to revise the title, since it contains repetitions (laparoscopic, safe), but the anatomical zone of the typology is not indicated. Response: The title has been revised.
  • Comment 2: In the "Keywords" "cholecystectomy" is repeated twice.     Response: Keywords have been amended.         
  • Comment 3: The phrase "anatomy of cholecystectomy" used in the "Keywords" and the text of the article seems incorrect to me.                                                   Response: It has been replaced by the phrase: <<surgical anatomy   of cholecystectomy>>                                                                 
  • Comment 4: There is a typo in line 152: instead of the number 65 it should be 64.     Response:    It has been corrected.    
  • Comment 5: After "Conclusions" you have not filled in lines 490 and 493-509.       Response:    Relevant lines have been completed.    

Reviewer 2 Report

Comments and Suggestions for Authors

Our surgical colleagues felt the need to reconsider the anatomy of Calot's triangle, given that during laparoscopic cholecystectomy, but I would also add open, lesions can be caused to both the right hepatic artery and the common bile duct and the hepatic ducts. The anatomy that they have studied and exposed well with the relative variations is explained perfectly and the paper can be an excellent read for all generations of doctors, young and old who often face this operation with the lightness of "silly" wisdom. We cannot criticize anything in the paper, we can only ask that it be concentrated in the length that in certain circumstances makes it verbose. We must ask that a few words be added on how the anatomy changes when we are faced with scleroatrophic cholecystopathies. In these cases in fact a medialization of lateral structures is found and vice versa. The cystic duct, together with the lower wall of the gallbladder and the common bile duct, can become a single body due to repeated inflammatory episodes caused by bacterial contamination or by stones that have passed into the main bile duct and have been removed with retrograde cholangiography. With enormous difficulty in separating the various components. But in sporadic cases, the portal branches themselves can be transposed to the lateral margin of the gallbladder bed, so that their detachment can lead to their injury. Not to mention, lastly, the porcelain gallbladder, which also creates difficulties due to the possible presence of a neoplasm and/or the difficulty in grasping it (doi.org/10.1007/s10353-021-00710-2 to be cited). Gallbladder surgery, although apparently simple, always offers the possibility of unpleasant surprises and this sentence should be the beginning of every study that aims to shed light on the subject. Compliments to the authors for the iconography, good English, the bibliography is an excellent basis for the paper

Author Response

  • Comment 1: Our surgical colleagues felt the need to reconsider the anatomy of Calot's triangle, given that during laparoscopic cholecystectomy, but I would also add open, lesions can be caused to both the right hepatic artery and the common bile duct and the hepatic ducts. Response: We totally agree with the reviewer 's point. We have commented on the text (abstract and main body) on the potential benefits of our proposal for safe laparoscopic and open cholecystectomy.
  • Comment 2: we can only ask that it be concentrated in the length that in certain circumstances makes it verbose.                                                                     Response:  A significant effort has been initially made to make the text concentrated and functional. We are afraid that, the messages this paper would like to give, may be affected if we do further modifications.
  • Comment 3: . We must ask that a few words be added on how the anatomy changes when we are faced with scleroatrophic cholecystopathies. In these cases in fact a medialization of lateral structures is found and vice versa. The cystic duct, together with the lower wall of the gallbladder and the common bile duct, can become a single body due to repeated inflammatory episodes caused by bacterial contamination or by stones that have passed into the main bile duct and have been removed with retrograde cholangiography. With enormous difficulty in separating the various components. But in sporadic cases, the portal branches themselves can be transposed to the lateral margin of the gallbladder bed, so that their detachment can lead to their injury. Response:  Although our study was performed in uncomplicated   cases of gallstone disease,  a comment on the difficulty in recognizing the anatomy of cholecystectomy properly in cases of cholecystitis and fibrosis has been included in the <<limitations of the study>> section.
  • Comment 4: Not to mention, lastly, the porcelain gallbladder, which also creates difficulties due to the possible presence of a neoplasm and/or the difficulty in grasping it (doi.org/10.1007/s10353-021-00710-2 to be cited).                           Response:  This important information has been incorporated in the text in the <<limitations of the study>> section. Unfortunately, we could not find in the pub-med  the above reference.

Comment 5:  Gallbladder surgery, although apparently simple, always offers the possibility of unpleasant surprises and this sentence should be the beginning of every study that aims to shed light on the subject.   Response:     We agree with the reviewer's point. An appropriate comment has been made in the study's abstract and Discussion section.                                                                        

Reviewer 3 Report

Comments and Suggestions for Authors

1.     Limited Generalizability Due to Sample Restrictions: While the study sample is robust, it consists only of patients with uncomplicated gallstone disease, excluding cases with inflammation or fibrosis. This limits the generalizability of the findings to more complex cases with severe acute or chronic cholecystitis, which often pose more significant challenges in anatomy visualization.

2.     Potential Complexity in Practical Application: The cognitive framework, though comprehensive, may be complex for some practitioners to implement, especially in high-pressure or emergency scenarios. This complexity might reduce its practical applicability without extensive training and experience.

3.     Absence of Intraoperative Validation Techniques: The study does not employ intraoperative cholangiography, which could have provided additional validation and safety confirmation in cases with complex anatomy or unexpected variations.

4.     Reliance on Michels' Landmark Studies: Although Michels' work is foundational, relying heavily on older anatomical studies may overlook recent advancements or alternative perspectives in laparoscopic anatomy research, which could add to the study’s contemporary relevance.

5.     Recent studies have explored using 3D imaging techniques (e.g., 3D angiography) to examine anatomical variations in the cystic artery. Wang et al. and Sugita et al. used preoperative imaging to identify abnormal distributions of the cystic artery, a technique similar to the one in this study for anatomical structure confirmation but with added predictive analysis before surgery.

6.     This study shares similarities with existing literature, particularly in the context of the critical view of safety, anatomical variation classification, and surgical training that may benefit from incorporating these studies' imaging techniques or combining predictive tools preoperatively to further enhance surgical safety and anatomical identification accuracy.

Author Response

  • Comment 1: Limited Generalizability Due to Sample Restrictions: While the study sample is robust, it consists only of patients with uncomplicated gallstone disease, excluding cases with inflammation or fibrosis. This limits the generalizability of the findings to more complex cases with severe acute or chronic cholecystitis, which often pose more significant challenges in anatomy visualization. Response: We agree with the reviewer’s comments. This is included now in the <<limitations of the study>> section.

  • Comment 2: Potential Complexity in Practical Application: The cognitive framework, though comprehensive, may be complex for some practitioners to implement, especially in high-pressure or emergency scenarios. This complexity might reduce its practical applicability without extensive training and experience. Response: We agree with the reviewer’s comments. The potential complexity in practical application is  mentioned now   in the <<limitations of the study>> section.

  • Comment 3: Absence of Intraoperative Validation Techniques: The study does not employ intraoperative cholangiography, which could have provided additional validation and safety confirmation in cases with complex anatomy or unexpected variations                                              Response:  the aim of our study was to form a common anatomic language combined with cognitive anatomical schemata that can be used as an anatomical map, helping the Surgeon to handle with confidence any variation that occurs in the process of achieving CVS. The use of intraoperative cholangiography was out of the scope of our study.

  • Comment 4: Reliance on Michels' Landmark Studies: Although Michels' work is foundational, relying heavily on older anatomical studies may overlook recent advancements or alternative perspectives in laparoscopic anatomy research, which could add to the study’s contemporary relevance. Response:      We used the two benchmark publications of Michels  as a glossary of anatomy to clarify and establish a terminology of the anatomic elements relevant to the procedure of cholecystectomy under the dictum of CVS. Also, extensive review of the literature was done.  The results were compared with those of a clinical study we perform of 279 patients undergoing LC for uncomplicated symptomatic gallstone disease.

  • Comment 5: Recent studies have explored using 3D imaging techniques (e.g., 3D angiography) to examine anatomical variations in the cystic artery. Wang et al. and Sugita et al. used preoperative imaging to identify abnormal distributions of the cystic artery, a technique similar to the one in this study for anatomical structure confirmation but with added predictive analysis before surgery. Response:  3D angiography, as well as MRCP, could be useful as a preoperative tool for the safe planning and execution of a difficult cholecystectomy. However ths was not the aim of our study. This comment is included in the   <<limitations of the study>> section.
  • Comment 6: This study shares similarities with existing literature, particularly in the context of the critical view of safety, anatomical variation classification, and surgical training that may benefit from incorporating these studies' imaging techniques or combining predictive tools preoperatively to further enhance surgical safety and anatomical identification accuracy.

            Response:   We think that our study expands the results of the existing literature

            regarding the anatomy in laparoscopic cholecystectomy in three aspects: (a) based on  

              the studies of Michels and  for the first time, as far we know, we gave a clear

          description and prevelance of the deep cystic artery which can have a separate course

          along the liver bed and outside the confine of the gallbladder wall, representing an

          anatomical structure with a great deal of confusion in various studies, as we pointed out

          in our text (b) we also focused on the structures beneath the cystic plate an anatomical

         field underreported up to now, and finally (c) we connected the anatomical structures

          found in certain surgical working spaces to the three components of the critical view of

         safety illuminating by this the interplay between the anatomy and the concept of the

          critical view of safety. We also pointed out that there is always a possibility of a third

          structure with a certain identity and characteristics that can be found during the effort to

         accomplish the critical view of safety.  

        We agree that our study findings can be combined with the information provided by

        other studies and create a more complete guide for the performance of safe

        cholecystectomy. This can be done by a future study.  

Round 2

Reviewer 3 Report

Comments and Suggestions for Authors

The author has revised the manuscript according to the suggestions.

  4o